# Bioaccessibility of Betalains in Beetroot (*Beta vulgaris* L.*)* Juice under Different High-Pressure Techniques

**DOI:** 10.3390/molecules27207093

**Published:** 2022-10-20

**Authors:** Urszula Trych, Magdalena Buniowska-Olejnik, Krystian Marszałek

**Affiliations:** 1Department of Fruit and Vegetable Product Technology, Prof. Waclaw Dabrowski Institute of Agricultural and Food Biotechnology—State Research Institute, 36 Rakowiecka St., 02-532 Warsaw, Poland; 2Department of Dairy Technology, Institute of Food Technology and Nutrition, University of Rzeszow, Ćwiklinskiej 2D St., 35-601 Rzeszow, Poland; 3Department of Food Technology and Human Nutrition, Institute of Food Technology and Nutrition, University of Rzeszow, 2D Zelwerowicza St., 35-601 Rzeszow, Poland

**Keywords:** bioaccessibility, high hydrostatic pressure, supercritical carbon dioxide, beetroot, betalains, antioxidant capacity, in vitro digestion model

## Abstract

The influence of high hydrostatic pressure (HHP) and supercritical carbon dioxide (SCCD) on the bioaccessibility of betalains in beetroot (*Beta vulgaris* L.) juice was investigated. Freshly squeezed juice (FJ) was treated at a mild temperature of 45 °C for 10 min (T45), pasteurization at 85 °C for 10 min (T85), HHP at 200, 400, and 500 MPa at 20 °C for 5 min (HHP200, HHP400, HHP500) and SCCD at 10, 30 and 60 MPa at 45 °C for 10 min (SCCD10, SCCD30, SCCD60). The juice was subjected to an in vitro digestion system equipped with dialysis. The content of betalains was measured with the aid of a High-Performance Liquid Chromatography (HPLC), the antioxidant capacity (AC) (ABTS•+, DPPH•) was analyzed during each digestion step, and the bioaccessibility of betacyanins and betaxanthins was assessed. The SCCD at 30 and 60 MPa significantly increased pigments’ bioaccessibility compared with other samples. The 30 MPa proved particularly advantageous, as it increased the bioaccessibility of the total betacyanins and the betaxanthins by 58% and 64%, respectively, compared to the T85 samples. Additionally, higher bioaccessibility of betacyanins was noted in HHP200 and HHP400, by 35% and 32%, respectively, compared to FJ, T45, and T85 samples. AC measured by ABTS•+ and DPPH• assays were not unequivocal. However, both assays showed significantly higher AC in SCCD60 compared to T85 (21% and 31%, respectively). This research contributed to the extended use of the HHP and/or SCCD to design food with higher health-promoting potentials.

## 1. Introduction

Beetroots (*Beta vulgaris* L.) are vegetables extremely rich in bioactive compounds with numerous pro-health properties [1,2], used in producing juices, ready-to-eat, frozen, or dried products, and are also used to obtain natural red food pigments. The leading group of phenolic compounds in beetroots are betalains as heterocyclic and water-soluble nitrogen-containing tyrosine-derived pigments with two subgroups: betaxanthins, yellow-orange pigments, and betacyanins responsible for red-purple color. The basis of the structure of these compounds is betalamic acid [4-(2-oxoethylidene)-1,2,3,4-tetrahydropyridine-2,6-dicarboxylic acid], derived from the cleavage of the aromatic ring of dihydroxyphenylalanine (DOPA) [3,4]. Red beets are among the top ten vegetables with the greatest antioxidant properties. Their consumption is recommended in order to prevent cancer and cardiovascular diseases [5,6]. Numerous studies reported health benefits such as strong anti-inflammatory and antibacterial effects, lowering blood pressure, and reducing low-density lipoprotein (LDL) levels [7,8,9,10]. Betalains are sensitive to pH changes, heat, light, and oxygen. Therefore, the matter of interest is the investigation of emerging food processing techniques to preserve betalains-rich products and improve the absorption of these pigments in the digestive system [8].

High Hydrostatic Pressure (HHP) is a food processing technique considered an alternative to thermal preservation, which is successfully used to damage pathogenic and spoilage microflora and reduce the activity of tissue enzymes responsible for browning reactions. In addition, it can assist the preservation of thermolabile bioactive ingredients, such as antioxidants, and retains sensory properties similar to the fresh product [11,12]. Supercritical Carbon Dioxide (SCCD) is a technique used in industry to extract selected compounds (caffeine, hops extracts, essential oils). However, this technique could be proposed as a good alternative for HHP and thermal pasteurization [13]. It can inactivate native microflora and significantly decrease tissue enzymes while the bioactive compounds are remained untouched. Due to the penetration of supercritical carbon dioxide into the sample, the tissues are physically disrupted, the cell membranes are modified, and the pH inside the cells is reduced. Therefore, the intracellular compounds could be extracted easily [14,15,16]. Depending on the process parameters, high pressure may increase, decrease or completely inhibit the activity of tissue enzymes. The decrease in enzyme activity is directly proportional to the pressure applied, and these reactions are most often described according to the first-order kinetic reactions [17]. The main limitation in using the HHP and SCCD techniques for preserving fruit and vegetable products is still incomplete knowledge of the parameters ensuring full inactivation of endogenous enzymes with high bar resistance. In addition, products are HHP treated in plastic packaging, and applying this technique in a continuous system is difficult. The SCCD treatment can only be used to preserve liquid foods, such as fruit and vegetable juices, beer, and milk, as processing the whole fruit causes significant tissue damage [12,13].

The concept of “bioaccessibility” is defined as the fraction of components released from the food matrix in the gastrointestinal tract and then possible to be absorbed by the intestinal epithelium cells and metabolized [18]. The calculation of bioaccessibility is essential to assess the intake of vitamins and other bioactive compounds into the body. Analyzing the bioaccessibility of antioxidants in high-pressure processed fruit and vegetable products may help design functional food with increased pro-health values. The most accurate model for bioaccessibility is in vivo studies with the human body simulation. However, it is associated with bioethical commissions and the complications with the standardization and comparability of research. Therefore, in vitro studies are valuable in the initial assessment of bioaccessibility, allowing an insight into the processes that may occur due to pH conditions and gastrointestinal enzymes. Cell cultures (e.g., Caco-2), centrifugation, or semi-permeable dialysis membranes are most often used to assess the absorption of substances into the bloodstream [18,19].

Increased awareness of the impact of diet on preventing civilization diseases like obesity, cardiovascular diseases, and cancer additionally emphasizes the importance of efforts to improve the bioaccessibility of bioactive compounds. In the current in vitro investigation, it is assumed that processing of beetroot juice under selected SCCD and HHP parameters may improve the bioaccessibility of betalains and antioxidant properties of juice after gastrointestinal digestion. In this context, this is the first study regarding the influence of SCCD and HHP on the bioaccessibility of betalains in the beetroot juice matrix.

## 2. Results and Discussion

### 2.1. Effect of HHP and SCCD on Betalains Content in Comparison to Thermal Treatment

Betanin, isobetanin, and neobetanin from the betacyanins, and vulgaxanthin I and vulgaxanthin II from betaxanthins were identified as the most abundant pigments in beetroot juice. Several compounds from the betaxanthin group, which were not identified qualitatively, were also detected (Appendix A). The betacyanins content in all betalains was as high as 88%; 86% (±1%) was represented by betanins. Betaxanthins constituted only 12% of total betalains, whereas vulgaxanthin I was the most common compound from this group of pigments (77% ± 1%). The juice treatment with both non-thermal techniques had no significant effect on the betalains composition. On the other hand, the same techniques significantly influenced the concentration and stability of betalains during simulated digestion. The content of betacyanins (betanin, isobetanin, and neobetanin) in fresh beetroot juice was 893 mg/L (Figure 1), whereas the content of betaxanthins was only 128 mg/L (Figure 2). These data are supported by other studies where authors reported betalains concentration in the range 800–1300 mg/L of fresh beetroot juice, with the highest concentration of betaxanthins being 75–95% and the lowest betacyanins approximately 5–25%, depending on the beetroot variety [3,20].

The content of both groups of these bioactive compounds in the fresh juice (FJ) did not differ significantly from the samples heated at 45 °C (T45) and 85 °C (T85). Other investigations proved that mild heating at temperatures below 50 °C should not affect the betacyanins. Moreover, betacyanins were not degraded by the short heat treatment (up to 3 min) even at 80 °C. However, the thermal treatment of beetroot at 90 °C for 3 min contributed to a decrease in the concentration of betacyanins by 25%. In contrast, the extension of the duration of the process to 10 min did not affect further betalains degradation [21]. High temperatures cause betanin degradation into betalamic acid and cyclo-DOPA. Isobetanin may also be found as a product of this degradation reaction. This hypothesis is very likely, as a slight increase in its content was noted in the T85 samples (5%). Most of the thermal reactions of degradation of betanin are reversible, depending on the initial betanin content, temperature, and acidity of the environment. The acidic environment favors the regeneration of betanin to its original form [21,22].

Similarly, to thermal treatment, HHP at 200 MPa and SCCD at 60 MPa were able to maintain the content of betacyanins and betaxanthins at a not statistically different level from fresh juice samples. Employing HHP at 400 and 500 MPa and SCCD at 10 and 30 MPa significantly affected the betalain content in beetroot juice compared to the other treatment parameters in both techniques. Both betacyanins and betaxanthins were equally sensitive to the aforementioned high-pressure processing parameters. The content of these pigments decreased in samples HHP400, HHP500, SCCD10, and SCCD30 compared to the FJ samples by 16%, 23%, 21%, and 34%, respectively. The lowest content of these compounds was recorded in SCCD30 samples. 

Depending on the applied parameters of HHP, other researchers noted both decreases [23,24] and increases [25,26] in betalain content. Sokołowska et al. (2017) [23] noted a decrease in betacyanins by 11.3–12.2% and betaxanthins by 7.7–8.9% in beetroot juice after application of high pressure at 300, 400, and 500 MPa for 10 min and the temperature up to 26 °C. It was found that betaxanthins were also more stable under HHP than betacyanins [23]. There is a hypothesis that higher degradation of these compounds may be caused by a baro-induced increase in oxygen in the sample, which may result in a decrease in the betalains content caused by faster oxidation processes. This hypothesis was confirmed by other authors who used HHP at 650 MPa, for at least 15 min. Moreover, they noted that prolonging the processing time from 15 to 30 min increased the betanins content compared to the control and blanched samples [25].

Considering HHP, the higher pressure resulted in the greater loss of betalain, unlike in SCCD, where the content of betalain in the SCCD60 samples was similar to the FJ samples. This phenomenon can be explained by the fact that the penetration of samples by supercritical CO_2_ under 10 and 30 MPa contributed to the reduction in free forms of betalains. At the same time, 60 MPa caused the release of betalains from the plant vacuoles and bounded in with proteins, fiber, or other carbohydrates. The obtained results indicate that the extraction of these compounds exceeded their degradation in the case of the highest-pressure parameters applied in the SCCD treatment of not from concentrate (NFC) beetroot juice. Higher pressures in the SCCD treatment contribute to a greater inactivation of tissue enzymes, which also prevents further degradation of bioactive compounds [3,4,14].

Liu et al. (2010) [27] analyzed the influence of SCCD in a continuous system in the pressure range of 4.5–30 MPa on, inter alia, the stability of betalains in red beet extracts. Continuous SCCD treatment at 30 MPa resulted in a significant decrease in betanin and isobetanin concentration by about 20%, similar to the present study [27]. In previous studies of the same research group, the content of betalains in the beetroot extract was examined after applying SCCD in a batch system at 37.5 MPa and 55 °C for 60 min or pasteurization at 95 °C and 5 min. The authors observed that SCCD treatment significantly decreased betanin, isobetanin, vulgaxanthin I, and II content by 7, 13, 17, and 19%, respectively. The thermal treatment not only resulted in a slight degradation of betacyanins but also increased the concentration of betaxanthins by about 14%. The spectrophotometric color measurement showed a greater proportion of yellow pigments in the blanched extracts, which was consistent with the increase in betaxanthins content and confirmed that these compounds might have been formed from other compounds under thermal treatment [28]. Studies on the effect of SCCD and thermal treatment on the degradation of betanin and isobetanin in aqueous solutions have shown that the degradation of these compounds follows first-order kinetic reactions. SCCD combined with a temperature of 45 °C (up to 66 °C) accelerated the degradation of betalains compared to thermal treatment alone [29]. Marszałek et al. (2017) [17] noticed a reduction in the betalains in beetroot juice after SCCD processing, along with increasing pressure, time, and temperature parameters. Under the most demanding conditions (60 MPa, 55 °C, 30 min), the degradation of betacyanins and betaxanthins reached 58% and 32%, respectively [17].

### 2.2. Effect of Simulated Gastrointestinal Digestion on Betalains Content in Beetroot Juice after Different Treatment

The decrease in betacyanins by 22% (±5%) during the simulated oral digestion was noted in all samples except the SCCD30. In the gastric digestion stage, the content of these compounds remained at a similar level, whereas a significant decrease was noted after the intestinal digestion step (the smallest decrease by 44% in the SCCD60 and 74% in the T85 samples, compared to the gastric digestion stage). Regarding betaxanthins, the conditions prevailing at the stage of simulated mouth digestion did not adversely influence the stability of these compounds. However, at the gastric digestion step, a significant decrease was observed (av. 70% ± 2%) compared to the oral digestion stage. As a result of simulated intestinal digestion, betaxanthins were further degraded by 17% (±2%), except for SCCD30 and SCCD60 samples (higher content of these compounds was noticed). Betaxanthins are more sensitive to an acidic environment than betacyanins, which have been mentioned in other studies [21,30]. Concerning betacyanin, an average of 46% (±3%) of the compounds passed the membranes during the dialysis step.

The highest concentration of betacyanins in the intestinal phase and dialysate was recorded in SCCD30, SCCD60, and HHP200 samples. It was reported that about 56% (± 3%) of initial concentration betaxanthins were detected in dialysate, whereas the highest content was detected in SCCD30 and SCCD60 samples. It means that SCCD technology improves the bioaccessibility of betaxanthins significantly. At the same time, the lowest content of betacyanins was noticed in thermally treated samples T45, T85, and HHP500. Moreover, the lowest content of betaxanthins was noted in HHP500 samples. Da Silva et al. (2019) [31] noted a gradual decrease in the betanin content after digesting the purified beetroot juice extract by 7% in the simulated oral cavity, 35% in the stomach, and 46% in the intestines, compared to the initial sample [31]. A purified extract could have contributed to greater exposure of betanins to acidic pH and a greater decrease in betanins content in the gastric stage compared to current findings with NFC (not from concentrate) juice. This phenomenon can be justified by a much higher concentration of other compounds like fiber in NFC juice compared to juice extracts. In the studies by Tesoriere et al. (2008) [30] on fresh and processed cactus pear fruit and beetroot and purified extracts, a protective impact of the matrix on the stability of betacyanins in simulated gastric digestion was demonstrated. In contrast, the matrix did not affect the stability of vulgaxanthin I, which was gradually degraded due to simulated digestion [30]. In the present study, the decrease in betacyanin and betaxanthin content after gastrointestinal digestion was similar among the samples treated in the same way. The loss of those pigments reached approximately 72% (±5%) in the HHP samples and approximately 79% (±1.5%) in the FJ, T45 and T85 samples compared to the initial content. In the SCCD samples, both pigments’ content decreased due to digestion by approximately 65% (±3), except in the SCCD30 sample, where betaxantines decreased was only 41% and betacyanins by 53%. Other studies regarding in vitro gastrointestinal digestion of beetroot juice reported a 96% loss of betacyanin and 73% loss of betaxanthin [32] as well as a 98.6% decrease in the total betalain content [33]. In the gastrointestinal tract, betalains can undergo decarboxylation, isomerization, or cleavage [32]. As their major metabolites, the betalain precursors (betalamic acid and cyclo-DOPA), 17-decarboxy-neobetanin, and 6′-*O*-feruloyl betanin were identified in the in vitro study [33]. Betalains are sensitive to light, oxygen, the activity of tissue enzymes, and high temperatures. They are also the most stable at pH 3–7. Their stability and extraction also depend on the structure and matrix of the tissue as well as the concentration of pigments [4].

### 2.3. Bioaccessibility of Betalains after HHP and SCCD Treatment in Comparison to Thermal Treatment

The bioaccessibility (BAc) of the total betacyanins ranged from 9.5% in the T45 and T85 samples to 22.6% in the SCCD30 samples. The BAc of total betaxanthins ranged from 11.5% and 11.8% in HHP500 and T85, respectively, to 32.3% in SCCD30 (Figure 3). Despite the higher initial stability of betaxanthins in fresh juice and thermal treated samples, the highest losses due to digestion were noted. The SCCD30 samples showed the highest BAc of betacyanin and betaxanthins despite the initially lowest stability of these compounds in the samples before digestion. Higher BAc of betacyanin and betaxanthins also distinguished the samples subjected to SCCD at 10 and 60 MPa than the others. The use of HHP improved the BAc of betacyanin compared to FJ, T45, and T85 but did not improve the BAc of betaxanthins. Among the HHP samples, 200 MPa had the most beneficial effect on the BAc of both groups of betalains.

No similar studies were conducted about HHP and SCCD on the BAc of betalains pigments in red beetroot juices. There is only one study on the effects of HHP on BAc of betalains from prickly pear fruit [34], as well as scarce studies on the effects of other emerging techniques on betalains BAc [35]. Other reports consider the effect of high-pressure processing on the BAc of different hydrophilic bioactive ingredients, including previous research by our team [36,37,38,39]. The BAc of betacyanins and betaxanthins varies due to differences in their chemical structure. Desseva et al. (2020) [32] reported a significantly lower recovery of betacyanins (approx. 4%) compared to betaxanthins (approx. 27%) in in vitro gastrointestinal digestion. The BAc of betalains also depends on the glycosylation of these compounds and accompanying components in the diet. In this group of pigments, betanin and betanidin are compounds with the highest antioxidant properties. Consumption of even tiny amounts of these compounds (in micromoles) by humans and animals reduces the oxidation processes of the lipid layer of cell membranes and inhibits heme decomposition processes in hemoglobin, myoglobin, and cytochromes [21]. The intestinal absorption of betalains is primarily mediated by direct diffusion. The concentration of betacyanins in the blood serum is low concerning the consumed dose, suggesting impaired absorption of these compounds. Some betalain digestion products are transported by protein conveyors, with energy expenditure. It is assumed that betacyanins are absorbed into the bloodstream unchanged and are not conjugated with glucuronic acid or sulphates [21,40].

In in vivo studies on rats administered intragastrically fermented beetroot juice found that both native and metabolized betacyanins can be absorbed through the gastric mucosa cells. Betanin, isobetanin, neobetanin, betanidines, and decarboxylated forms of betacyanins were largely detected in the gastric fluids and the blood and urine of rats [40]. In studies involving 10 healthy men who consumed beetroot juice or whole beets, no betanin was detected in the blood plasma. However, NOx levels increased 8 h after juice consumption [41].

Gomez-Maqueo et al. (2021) [34] demonstrated a positive impact of HHP at 350 MPa and 5 min on the BAc of betanin from prickly pear fruit pulp. Depending on the fruit variety, an increase of 20–27% was noted compared to non-treated samples. On the other hand, there was no improvement in the BAc of indicaxanthin and peel-derived betalain. Using parameters 100 and 600 MPa simultaneously did not bring positive results. The BAc was calculated from the non-treated samples to the samples centrifuged after intestinal digestion. Similar to our research, the effect of pressure parameters on BAc was irrespective of the initial content of these compounds before digestion. These results were justified by an increase in the extractability of bioactive compounds from plant tissue and changes in the components of the fruit matrix, such as soluble dietary fiber, which could increase the stability of betalains in the digestive system [34]. The HHP treatment can reduce the amount of polyphenol complexes with proteins compared to thermally treated samples, which may increase the BAc of polyphenols [36]. Our previous research focused on the influence of HHP and SCCD treatment on the BAc of anthocyanins and vitamin C in blackcurrant juice/puree, indicating that both techniques can improve the BAc of bioactive compounds, depending on the parameters used. The improvement of bioaccessibility as a result of different high-pressure processing methods may occur due to increased extraction of bioactive compounds from the tissue as well as binding them with macromolecular compounds such as fiber or pectin. The metabolites formed during digestion significantly impact the products’ antioxidant properties [37,38].

### 2.4. The Influence of Treatment on the Antioxidant Properties of Beetroot Juice in an In Vitro Model of the Gastrointestinal Digestion 

The antioxidant capacity (AC) in the non-digested beetroot juice, according to analysis using DPPH• radicals, was the highest in HHP200 and FJ samples (about 7.6 µM/mL) (Figure 4). The application of other techniques (thermal, HHP, or SCCD) contributed to the reduction in the AC of beetroot juice compared to FJ samples. The lowest AC measured by the DPPH• assay was noted in HHP400 samples (5.00 µM/mL). After in vitro oral digestion, a notable AC reduction in all samples ranging from 37% in HHP400 to 74% in T85 was noticed. On the other hand, under the conditions prevailing during simulated gastric digestion, an increase in AC was remarked in all samples by 56% (±10%) compared to the previous digestion step. In the samples treated by HHP, the AC reached a value greater than before digestion, which may indicate the release of metabolites of compounds with high AC. After digestion in the intestine conditions, the AC measured by the DPPH• assay decreased by 66% (±7%). The highest AC in the dialysate was achieved by SCCD30 and SCCD60 samples (2.74 and 2.78 µM/mL, respectively), and the lowest by FJ samples (1.65 µM/mL). The AC of the SCCD10 and HHP500 samples was statistically significantly higher than those treated at 400 and 200 MPa and heat treated at 45 °C and 85 °C. In the DPPH• assay, the correlation between the AC and the total betalains in the beetroot juice was noticeable.

The AC in the control samples measured by the ABTS•+ assay was the highest in SCCD60 (9.81 µM/mL) and was not statistically significantly different from SCCD10 and HHP200 (Figure 5). The samples after thermal treatment at 85 °C, HHP400, HHP500, and SCCD30 had lower AC than the FJ samples. The lowest AC was recorded in HHP500 samples (7.41 µM/mL). The ABTS•+ assay did not indicate a decline in AC after digestion with the salivary enzymes and oral pH. Contrary to the DPPH• method, there was a decline in AC caused by simulated gastric digestion compared to the previous digestion step, from 15% in T85 to 37% in SCCD60 samples. The exception is the AC of the HHP200 sample, which increased by 9% compared to the oral digestion stage while reaching the highest value. Simulated intestinal digestion increased the AC in all samples compared to gastric digestion. In addition, dialyzed samples reached higher AC than before dialyze, except for samples after SCCD treatment, where the AC remained at a similar level. This might be caused by the attendance of compounds characterized by high antioxidant activity in the dialysate. Despite the decreasing stability of betalains due to unfavorable conditions in the gastrointestinal tract, their antioxidant properties have not been lost, as the metabolites of these compounds are still distinguished by high antioxidant activity. Following the ABTS•+ assay, the highest AC in the dialysate was found in the SCCD60 samples (10.87 µM/mL). They were not statistically significantly different from the AC of the FJ, T45, and HHP200 samples. The lowest AC was recorded in HHP500 samples (7.96 µM/mL). The results of the AC obtained with ABTS•+ and DPPH• methods are not unequivocal.

The beneficial effect of HHP and SCCD treatment on the antioxidant properties of fruit and vegetable products was also noticed by previous investiagtions [42,43,44]. Gómez-Maqueo et al., 2019 [42] reported a significant improvement in the antioxidant activity of the pulp of the prickly pear after using the HHP at 350 MPa/5 min. It was correlated with the content of phenolic compounds. Rodríguez-Roque et al. (2015) [43] pointed to the possibility of improving the antioxidant capacity (DPPH•) in fruit drinks with the addition of milk by treatment with HHP (400 MPa, 5 min). In addition, this treatment allowed the preservation of the higher AC of the fruit juices compared to the thermal processing, regardless of the matrix. Briones-Labarca et al. (2011) [44] also reported stronger antioxidant properties of apples processed with HHP (500 MPa, 2–10 min) compared to untreated samples. 

Our team’s research has reported a positive effect of the SCCD at 30 and 60 MPa on the AC (ABTS•+ and DPPH•) of blackcurrant juice subjected to simulated gastro-intestinal digestion compared to thermally treated juices (45 and 85 °C, 10 min). SCCD improved the availability of antioxidants for digestion, thereby releasing more products of metabolism with potent antioxidant properties, such as protocatechuic acid [38]. Wang et al., 2020 [22] obtained similar AC results in the digestion stages as in DPPH• tests in this study. Under the oral digestion conditions, there was a decrease in AC. However, it was reversible at the gastric digestion step and rich the highest AC in those conditions. The decrease in AC after intestinal digestion occurred in the DPPH•, ABTS•+, and FRAP (ferric-reducing antioxidant power) assays. The loss of antioxidant activity in beetroot after heat treatment due to the loss of phenols, flavonoids, and betalains was also indicated [22]. Another study compared the AC of 23 fruits and vegetable juices bought in the local market before and after in vitro digestion. The beetroot juice is distinguished by the highest content of total polyphenols and AC, among other juices. According to the FRAP assay, beetroot juices from both producers showed a two-fold increase in AC at the stomach digestion stage. In the first juice, which also showed a slight decrease in total polyphenols (6%), this high level of AC was maintained even after intestinal digestion. In the second juice, the AC returned to a value close to that obtained in the fresh juice, although the polyphenol content was almost halved due to intestinal digestion. This supports the assumption that AC is shaped by metabolites formed upon digestion of the native antioxidants in beetroot. Less antioxidant losses than our study may be due to the omission of the digestive step in the mouth, the reduction in gastric digestion to 1 h, and the use of centrifugation after duodenum digestion, as opposed to dialysis, as in this study. As in the present study, the variation in AC was strongly related with the application of DPPH• or ABTS•+ radicals. This shows that the methods for determining activity do not always overlap, and the response to radicals may differ depending on the matrix [45].

Desseva et al. (2020) [32] also obtained varied results of AC analysis in beetroot juice after digestion by DPPH•, ABTS•+, FRAP, and CUPRAC (cupric ion reducing antioxidant capacity) assays. The authors report that the differences in results can be the presence of bile acids, the affinity of the radicals for the sample components, a different mechanism of radical formation and stability, and other reaction temperature conditions. Metabolites formed from the breakdown of betalains and contribute to the high antioxidant capacity of the digested juice are betalain deglycosylation products, as well as neobetanine or cyclo-DOPA. Betalamic acid and cyclo-DOPA glucoside formed in the alkaline environment do not exhibit antioxidant properties, but they can demonstrate them at acidic pH [21].

## 3. Materials and Methods

### 3.1. Reagents and Auxiliary Materials

Cellulose membrane for dialyzing (avg. flat width 25 mm, molecular weight cut-off = 14,000) and digestive enzymes such as α-amylase (TDF-100 A, 24,975 U/mL), mucin from the porcine stomach—type II, pepsin from the porcine gastric mucosa (250 U/mg solid), pancreatin from the porcine pancreas (8 × USP specifications), porcine bile extract, and reagents such as sodium dodecyl sulfate—ACS reagent, sodium bicarbonate ≥ 99.5%, (±)-6-hydroxy-2,5,7,8-tetramethyl-chromane-2-carboxylic acid (Trolox), 2,2-diphenyl-1-picrylhydrazyl (DPPH• radical), 2,2′-Azino-bis (3-ethylbenzothiazoline-6-sulfonic) acid, diammonium salt (ABTS+• radical), acetonitrile (HPLC) and sodium hydroxide pellets ≥ 98.0% (NaOH) were provided by Sigma-Aldrich (St. Louis, MO, USA).

Reagents such as disodium hydrogen phosphate anhydrous pure p.a. ≥ 99.0% (Na_2_HPO_4_), dipotassium hydrogen phosphate (K_2_HPO_4_), sodium chloride pure p.a. ≥ 99.9% (NaCl), and di-sodium wersenate standard solution 0.01 mol/L were supplied from Chempur (Piekary Śląskie, Poland). In addition, Honeywell Fluka (Seelze, Germany) provided pure hydrochloric acid p.a. ACS reagent 37% (HCl) and potassium peroxodisulfate ≥ 99.0%, and Avantor (Gliwice, Poland) provided formic acid 98–100% CZDA, ethanol 96% CZDA and methanol (HPLC grade).

### 3.2. Testing Material

Fresh beetroot (*Beta vulgaris* L.) juice was selected for the research, and vegetables of Polish origin were purchased at the local market. The vegetables were washed and squeezed in the Robot Coupe Juicer (model J80 Ultra, France) to produce beetroot juice. One batch of juices was left without any treatment (FJ). The next two batches of samples, in the same type of packaging, were subjected to thermal treatment at 45 °C for 10 min (T45) and pasteurization at 85 °C for 10 min (T85) in a pasteurization water bath (Labo Play, Bytom, Poland). Mild heat treatment was used to verify if the temperature was relevant in the SCCD, while pasteurization is a traditional method of juice preservation used as a comparative method for emerging techniques. SCCD juice treatment was carried out in 150 mL glass jars at three pressure parameters: 10, 30, and 60 MPa, for 10 min, at 45 °C (SCCD10, SCCD30, SCCD60) in the batch mode in a Speed SFE 4 (Applied Separations, Allentown, PA, USA). The HHP treatment was carried out in 250 mL PET bottles at 200, 400, and 500 MPa, 5 min, and 20 °C (HHP200, HHP400, HHP500) in the CALIBER 70 × 1 device (EXDIN Solutions Sp. z o.o., Kraków, Poland). In both methods of high-pressure treatment, the applied pressure conditions covered the whole scope of the parameters in the devices, with the optimal time length of the processes. Until the analyzes were performed, the samples were stored in a freezer at −20 °C. Controls are all sample types regardless of treatment but prior to simulated digestion. Appendix A in the Appendix A presents the research’s scope and workflow.

### 3.3. Simulation of Gastrointestinal Digestion and Determination of Bioaccessibility

In vitro, gastrointestinal digestion simulation among dialysis was carried out under the model presented by Buniowska et al. (2017) [46] and Minekus et al. (2014) [19]. Oral digestion was simulated by mixing 50 mL of the juice or water as a blank sample and 5 mL of salivary enzyme solution. Using HCl or NaOH buffers, the pH was adjusted (HI 211 m, Hanna Instruments, Woonsocket, RI, USA) to 6.75 ± 0.20, and then the solution was left for 10 min in a shaking water bath (Labo Play, SWB 8N, Bytom, Poland) at 37 °C, 90 rpm. After collecting part of the sample for analysis, 20 mL of pepsin solution (2 g of NaCl, 7 mL of HCl, 3.2 g of pepsin per 1 L of distilled water) was added, and the pH was adjusted to 2.00 as needed. Then, samples were incubated for 2 h again in the previously described parameters. To initiate simulated intestinal digestion, 20 mL of the solution from the gastric phase was moved to the clean bottle. Its pH was adjusted to 5.00 ± 0.20 and mixed with 5 mL of pancreatin (1 g/L) and bile solution (25 g/L). The cellulose membrane, cut into 30 cm sections, sterilized, washed, and filled with 25 mL of NaHCO_3_ (0.5 M, pH 7.5), was dipped in the prepared sample and incubated for 2 h. The solution inside the dialysis membrane is considered potentially absorbed into the bloodstream. The samples were inserted into an ice bath for 10 min to complete the digestion process. All kinds of juice samples were subjected to simulated digestion in triplicate. Samples of each digestion and dialysis step were collected and stored in the −20 °C freezer until analysis. Equation (1) was employed to determine bioaccessibility (BAc).
BAc (%) = 100 × (BC_digested_/BC_before digestion_)(1)

Equation (1). Calculating bioaccessibility (BAc—bioaccessibility of the analyzed compound; BC_digested_—the amount of analyzed compound in the juice after digestion; BC_before digestion_ —the amount of analyzed compound in the juice before digestion).

### 3.4. Chemical Analysis

#### 3.4.1. Identification of Betalains

Fresh beetroot juice was 5 times diluted with 0.2% formic acid (phase A) and then centrifuged. A total of 2 mL of the supernatant was applied to a Sep-Pak C18 column (Waters, Milford, CT, USA); The betaxanthins were eluted with 6 mL of phase A and the betacyanins with 6 mL of phase B (acetonitrile), both were collected in 10 mL flask and made up to the mark. The total concentration of both dyes present in the collected fractions was determined spectrophotometrically based on the molar extinction coefficients and calculated according to Equation (2).
Betalain content (mg/L) = A × DF × MV × 1000/(e × l)(2)

Equation (2). The concentration of betalain (A—absorption, DF—dilution factor, MW—molecular weights, e—molar extinction coefficient, l—pathlength. (Betacyanins—MW = 550 g/mol; e = 60,000 L/mol × cm; Betaxanthins—MW = 308 g/mol; e = 48,000 L/mol × cm) were applied.

The obtained fractions with known concentrations were used to prepare a series of dilutions for standard curves (R2 > 0.99). LOD and LOQ for both groups of pigments were determined on the basis of the signal to noise ratio, which was S/N = 10 in the case of LOQ, and S/N = 3 in the case of LOD. Betaxanthins: LOQ ≥ 0.5 mg/L; LOD ≥ 0.2 mg/L. Betacyanins LOQ ≥ 2 mg/L LOD ≥ 0.6 mg/L.

#### 3.4.2. HPLC Analyses of Betalains

The analysis was carried out by the methodology presented by Ravichandran et al. (2013) [47]. For this purpose, a Waters 2695 HPLC system connected to a Photodiode Array Detector 2996 (Waters, Milford, CT, USA) was used. A Sunfire C8 column (250 × 4.6 mm, 5 μm, Waters, Milford, CT, USA) and a pre-column (Sun Fire C8, 20 × 4.6 mm, 5 mm, Waters, USA) were heated to 30 °C. Samples of the juices were diluted as needed and filtered through the 0.45 μm syringe filter (Waters, Milford, CT, USA). During 60 min and the flow of 1 mL/min, 10 µL of the juice was dispensed with 0.2% formic acid (A) and acetonitrile (B). The flow gradient of the eluates was as follows: A: B [%]; 100:0 (0 min); 100:0 (0–7 min); 97:3 (7–17 min); 90:10 (17–27 min); 90:10 (27–35 min); 80:20 (35–45 min); 0:100 (45–50 min); 100:0 (50–55 min); 100:0 (55–60 min). Betacyanins were detected at 538 nm and expressed as betanin equivalent, while betaxanthins were detected at 480 nm and expressed as vulgaxanthin I equivalent.

Individual betalains were identified based on an earlier experiment [23] and literature data on the typical retention order during the RP-HPLC separation [48,49,50], as well as their fluorescent properties at an excitation wavelength of 465 nm and an emission wavelength of 510 nm. Yellow pigments are highly fluorescent because of four conjugated double bonds, while betacyanins do not present fluorescence due to the conjugation of bonds with the cyclo-DOPA aromatic ring [51].

#### 3.4.3. ABTS+• Radical Assay

To determine the antioxidant capacity with the ABTS+• radicals, the method presented by Re et al. (1999) [52] was applied. The cationic solution of radicals was prepared by mixing 7 mM ABTS+• and 2.45 mM potassium persulfate and leaving it in the darkness for 18 h before starting the analyses. The radical solution was diluted with ethanol to obtain an absorbance of 0.740–0.750 at a wavelength of 734 nm on a UV/Visible Spectrophotometer (Ultrospec 2000, Pharmacia Biotech, Amersham, UK). Trolox (1 mg/mL) in ethanol was applied to prepare a six-point calibration curve with the following concentrations: 50, 100, 150, 200, 250, and 300 µg/mL. Then, 0.025 mL of the juice, Trolox standard in the case of a blank sample, and 2.5 mL of ABTS+• solution was dispensed into the cuvette and incubated at 30 °C for 6 min. The absorbance was determined at 734 nm, and the result was expressed in Trolox Equivalent Antioxidant Capacity (TEAC).

#### 3.4.4. DPPH• Radical Assay

To determine the antioxidant capacity with the DPPH• radicals, the method presented by Yen and Chen (1995) [53] was used. The DPPH• radicals (1 mM) were dissolved in methanol, incubated for 3 h in the darkness, and diluted approximately 10-fold to obtain the absorbance of 0.700–0.800 at 517 nm on a spectrophotometer described in the previous assay. A standard curve was also made of a solution of Trolox in methanol (1 mg/mL), obtaining the concentrations in the following points: 10, 20, 30, 40, 50, and 100 µg/mL. The analysis was performed by dispensing 0.1 mL of juice, standard or distilled water sample as a blank and 2 mL of DPPH• radical solution into the cuvette, then leaving it in the darkness for 30 min and room temperature (22 ± 1 °C) to incubate. The absorbance was measured at 517 nm. The result was expressed in Trolox Equivalent Antioxidant Capacity (TEAC).

### 3.5. Statistical Analysis

The one-way analysis of variance was performed with the ANOVA test, and the significance analysis of differences in mean values with the Tukey’s test on the confidence level of α = 0.05. The Statistica 7.1 software (StatSoft, Tulsa, OK, USA) was employed for statistical analysis. Each type of beetroot juice sample was conducted into simulated digestion in triplicate, and chemical analyses were performed in duplicate.

## 4. Conclusions

Thermal treatment (45 and 85 °C) did not reduce the stability of betacyanin and betaxanthins in the control samples (non-digested). Applying HHP200 and SCCD60 also allowed for maintaining the high stability of betacyanin and betaxanthins. The type of the beetroot juice treatment and process parameters did not affect the betalain composition but influenced their stability during simulated digestion. As a result of the unfavorable pH conditions and the presence of enzymes, the content of betalains decreased during digestion.

In the juices subjected to SCCD at 30 and 60 MPa, the bioaccessibility of betacyanins and betaxanthins was statistically significantly higher than in all other types of samples. The 30 MPa proved particularly advantageous, as it increased the bioaccessibility of the total betacyanins and the total betaxanthins by 58% and 64%, respectively, concerning the pasteurized samples. In the juices treated with HHP at 200 and 400 MPa, a significant improvement in the bioaccessibility of betacyanins by 35% and 32%, respectively, was also observed in fresh juices and thermally treated. The AC in the beetroot juices differed depending on the radical used in the analysis. The decreased stomach AC in the stomach in the ABTS•+ assay correlated with the betalain concentration decrease at this step of simulated digestion. In turn, in the method with DPPH• radicals, there was a significant increase in AC in the stomach. This may indicate greater sensitivity to compounds stable at acidic pH (such as polyphenols and betacyanins). The DPPH• assay showed a decrease in the AC after digestion, which was noticeably higher in the SCCD and HHP500 than in the other samples. According to the ABTS•+ assay, the antioxidant capacity increased by 7–14%, and in dialysate, it was significantly higher in SCCD60, HHP200, FJ, and T45 samples compared to others.

Applying high-pressure techniques such as HHP and SCCD to beetroot juices with selected pressure parameters may improve the bioaccessibility of betalains. Increasing the extraction and affecting the food matrix could improve the stability of betalains in the gastrointestinal tract, making them more accessible for metabolism and absorption. It would be necessary to confirm the beneficial influence of high-pressure techniques on the bioaccessibility of betalains and the related health benefits for the human body in in vivo studies.

## Figures and Tables

**Figure 1 molecules-27-07093-f001:**
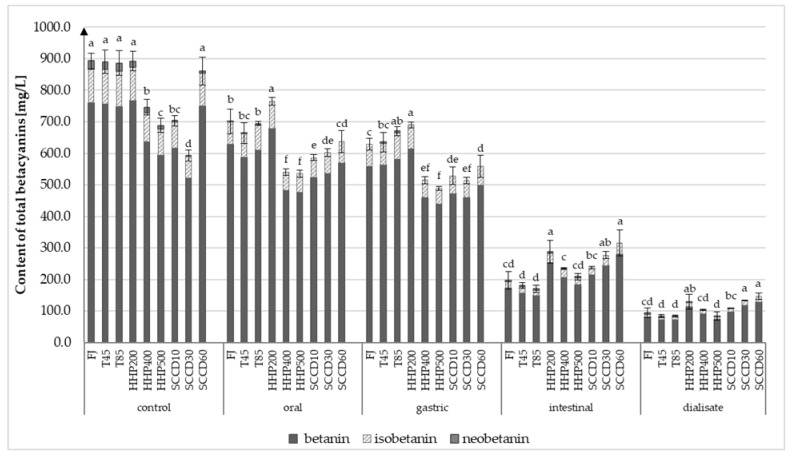
Content of total betacyanins (betanin, isobetanin, and neobetanin) in beetroot juice after different processing on every stage of simulated gastro-intestinal digestion. FJ—untreated juice; T45—thermal treatment of sample at 45 °C; T85—pasteurized sample at 85 °C; HHP200—sample subjected to HHP at 200 MPa; HHP400—sample subjected to HHP at 400 MPa; HHP500—sample subjected to HHP at 500 MPa; SCCD10—sample treated by SCCD at 10 MPa; SCCD30—sample treated by SCCD at 30 MPa; SCCD60—sample treated SCCD at 60 MPa. (Different letters over the bars indicate significant differences between total betacyanins content at each step of simulated digestion (*p* ≤ 0.05).

**Figure 2 molecules-27-07093-f002:**
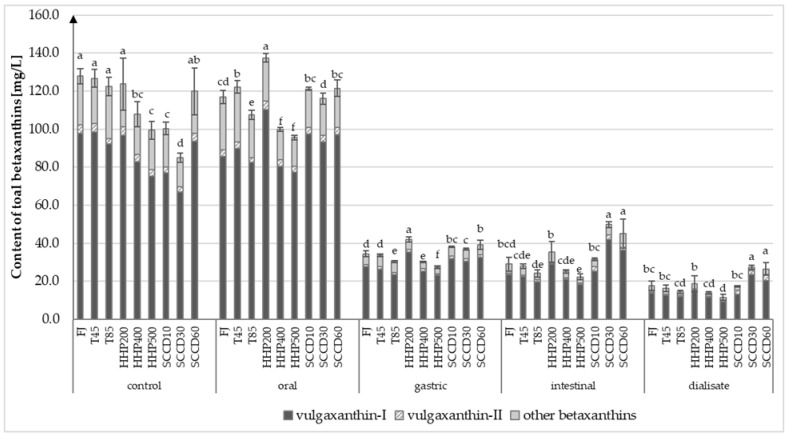
Content of total betaxanthins (vulgaxanthin-I, vulgaxanthin-II, and other betaxanthins) in beetroot juice after different processing on every stage of simulated gastro-intestinal digestion. T45—thermal treatment of sample at 45 °C; T85—pasteurized sample at 85 °C; HHP200—sample subjected to HHP at 200 MPa; HHP400—sample subjected to HHP at 400 MPa; HHP500—sample subjected to HHP at 500 MPa; SCCD10—sample treated by SCCD at 10 MPa; SCCD30—sample treated by SCCD at 30 MPa; SCCD60—sample treated by SCCD at 60 MPa. (Different letters over the bars indicate significant differences between total betaxanthins content at each step of simulated digestion (*p* ≤ 0.05).

**Figure 3 molecules-27-07093-f003:**
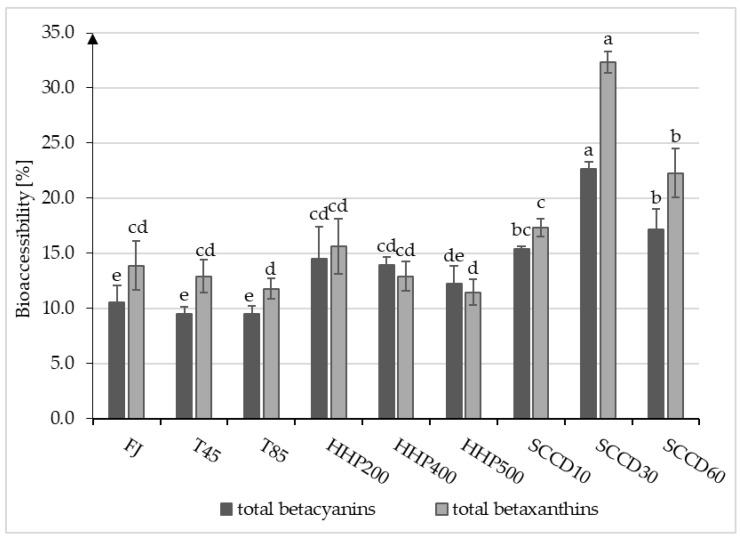
Bioaccessibility (BAc) of total betacyanins and total betaxathins from beetroot juice after different processing: T45—thermal treatment of sample at 45 °C; T85—pasteurized sample at 85 °C; HHP200—sample subjected to HHP at 200 MPa; HHP400—sample subjected to HHP at 400 MPa; HHP500—sample subjected to HHP at 500 MPa; SCCD10—sample treated by SCCD at 10 MPa; SCCD30—sample treated by SCCD at 30 MPa; SCCD60—sample treated by SCCD at 60 MPa. (Different letters over the bars indicate significant differences between the mean bioaccessibility of betacyanins and betaxanthins (*p* ≤ 0.05).

**Figure 4 molecules-27-07093-f004:**
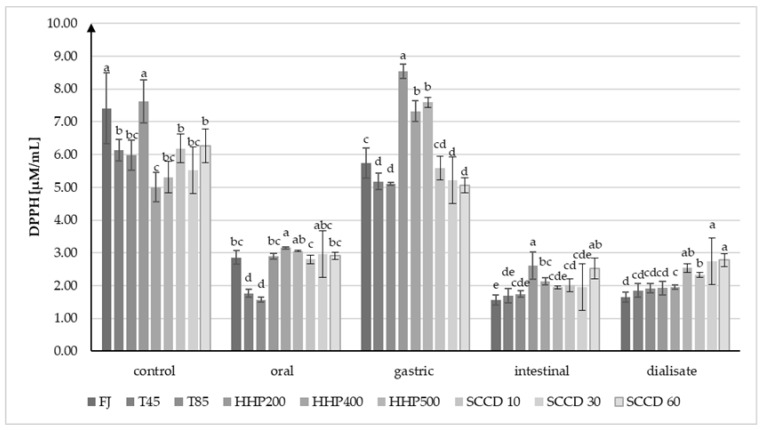
Results of DPPH• analysis of antioxidant properties (AC) of beetroot juice after different treatments on every stage of simulated digestion. FJ—untreated juice; T45—thermal treatment of sample at 45 °C; T85—pasteurized sample at 85 °C; HHP200—sample subjected to HHP at 200 MPa; HHP400—sample subjected to HHP at 400 MPa; HHP500—sample subjected to HHP at 500 MPa; SCCD10—sample treated by SCCD at 10 MPa; SCCD30—sample treated by SCCD at 30 MPa; SCCD60—sample treated by SCCD at 60 MPa. (Different letters over the bars indicate significant differences between the mean antioxidant capacity at each step of simulated digestion (*p* ≤ 0.05).

**Figure 5 molecules-27-07093-f005:**
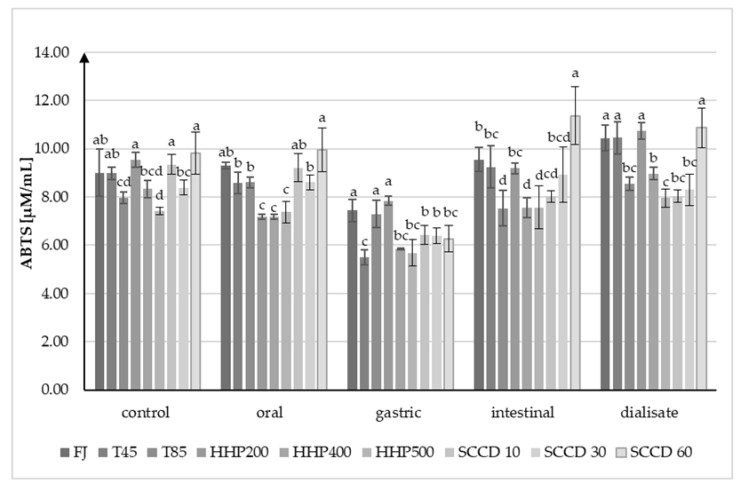
Results of ABTS•+ analysis of antioxidant properties (AC) of beetroot juice after different treatments on every stage of simulated digestion. FJ—untreated juice; T45—thermal treatment of sample at 45 °C; T85—pasteurized sample at 85 °C; HHP200—sample subjected to HHP at 200 MPa; HHP400—sample subjected to HHP at 400 MPa; HHP500—sample subjected to HHP at 500 MPa; SCCD10—sample treated by SCCD at 10 MPa; SCCD30—sample treated by SCCD at 30 MPa; SCCD60—sample treated by SCCD at 60 MPa. (Different letters over the bars indicate significant differences between the mean (*p* ≤ 0.05) antioxidant capacity at each step of simulated digestion).

## Data Availability

The original data presented in the study are included in the article, further inquiries can be directed to the corresponding author.

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
