# Peer review of "Bioaccessibility of Betalains in Beetroot (*Beta vulgaris* L.*)* Juice under Different High-Pressure Techniques"

_molecules, 2022, doi:10.3390/molecules27207093_

Round 1
Reviewer 1 Report
This manuscript reports the impact of high hydrostatic pressure and supercritical carbon dioxide processing on the bioaccessibility of betalain in beetroot juice after simulated gastrointestinal digestion, as well as changes in antioxidant activity. The results of these nonthermal techniques were also compared to those of thermal pasteurization treatments. The work is original and worthy of investigation and the experimental design seems adequate.
Main comment:
- It is an extensive laboratory work, with many samples analyzed and, therefore, many results. These are clearly presented, but have been compared with a very large number of other studies in the literature, which makes the manuscript quite extensive and a little less focused on its main findings. Therefore, it is suggested that the results be compared in a more succinct and summarized way.
Comments and suggestions:
Abstract:
- Replace “beetroot juice (Beta vulgaris L.)” with “beetroot (Beta vulgaris L.) juice”.
Introduction:
- “Numerous studies reported health benefits such as strong anti-inflammatory and antibacterial effects, blood pressure, and low-density lipoprotein (LDL) levels [7–10].” This sentence is not entirely clear. What are the effects on blood pressure and LDL levels?
- “Supercritical Carbon Dioxide (SCCD) is a technique used in industry to extract selected compounds (caffeine, hop, essential oils).” Hop is a plant, not a compound.
Results and discussion:
- Page 3. It is mentioned that “Several compounds from the betaxanthin group, which were not identified qualitatively, were also detected.” Therefore, it would be interesting to provide the HPLC chromatograms on supplementary material. So it would be possible to see the number of peaks and their intensity compared to those that were quantified.
- Page 5. Please check the sentence: “In contrast to the UHT treatment (131 °C/2 s), which caused significant losses of these compounds.” It looks incomplete or unfinished.
- Page 6. The figure is not numbered correctly. This should be figure 4 and not figure 2. However, it does not seem to be very relevant to present the chemical structure (and not “chemical formulas”) of these compounds in the manuscript as it is not discussed in the main text. Therefore, it may be moved to supplementary material or deleted from the manuscript.
- Subsection 2.1. discusses the “effect of HHP and SCCD on betalains content in comparison to thermal treatment”, the second paragraph in page 6 starts discussing the effect of simulated gastrointestinal digestion. Therefore, it will be necessary to add a title to this second subsection (2.2.).
- Page 7. The meaning of “NFC” must be indicated.
- Page 11. A comma is required: FJ, T45 and HHP200 samples.
Material and methods
- Page 12. The 4 in K2HPO4 must be subscribed.
- For the “HPLC analyses of betalains”, the quantification process should be better explained and the calibration curves should be presented, as well as LOD and LOQ values.
- A reference is needed for the following sentence: “Yellow pigments are highly fluorescent because of four conjugated double bonds, while betacyanins do not present fluorescence due to the conjugation of bonds with the cyclo-DOPA aromatic ring.”
- In the “DPPH• Radical Assay”, it is mentioned “To determine the antioxidant capacity of the DPPH• radicals”. However, it was not intended to evaluate the antioxidant activity of DPPH radicals, but of the beetroot samples under analysis.
Other observation:
- Use “min” or “minutes” throughout the manuscript. “min” is suggested.
Reviewer 2 Report
The paper is excellent. I only recommend the authors to rewrite Figure 6 with bars in color
Author Response
Dear Mrs/Mr Reviewer
On behalf of all the co-authors, we would like to thank you for your favorable opinion and kind words about our manuscript. We are very pleased that our effort put into preparing this study has been appreciated.
We are very grateful for the time you spend to review this article and your valuable suggestion on better presentation the results in a bar chart. We understand the point of your remark and agree that the colors on the bars would allow for a greater contrast between them. However, in that case we would have to change all charts in color versions and unfortunately we cannot provide this at the moment. In the chart you mentioned (according to the new numbering it is chart 4), the data is presented in order, which we believe is enough to appropriate reading.
Hopefully the presence of this slight imperfection will not prevent publication of this manuscript. Thank you again for contributing to this article.
Your sincerely,
Urszula Trych
Reviewer 3 Report
Reviewer's Report
To Authors:
A manuscript entitled "Bioaccessibility of betalains in beetroot (Beta vulgaris L.) juice under high-pressure techniques" was submitted for consideration for publication in Molecules.
In this study, the authors have investigated the effect of high-pressure techniques on different betalain-constituents of fresh and thermal-treated beetroot juice. The authors have used high hydrostatic pressure (HHP) and supercritical carbon dioxide at different conditions (temperature, time, pressure, etc.) before performing in vitro digestion. Higher bioaccessibility of betacyanins was noted in HHP200 and HHP400 samples. In general, all technics (except HHP200) contributed to the reduction of the AC of beetroot juice compared to fresh juice samples.
Specific comments and suggestions to the authors are described below
In general, the title is informative it reflects the paper content. I suggest to be included “different” between “under High-Pressure” or to write full name of both technics.
Abstract: In general, the abstract describe research and it is well-written. Numerical values for the most important findings should be included. Also, L5 – delete ‘and’
Introduction: In general, it provides a background and argumentation. Although, the aim of the study is formulated, I suggest formulation of a proper hypothesis of the study. Moreover, a better definition of betalains should be provided – you may use also nitrogen-containing ‘tyrosine-derived pigments’ instead of ‘phenols’. It would be good to give definition for ‘bioaccessibility’. Finally, along with the positive aspects of the techniques used, their disadvantages should also be noted in this section.
Materials & Methods: minor imperfection in this section:
p. 12 – write full name of EDTA correctly; p. 14 – 3.4.3. 10…………100 ? g/mL – micro? 3.5. Each type o beetroot juice… - of.
Results & Discussion:
I suggest Fig. 1 to be sent in Materials and Methods section or better as a Suppl. material.
p. 4 Fig. 2 (bar graph) – why SCCD60 results is close to the control while SCCD10 and 30 have lower values?
p. 6 Fig. 2 – sent as a Supplementary material.
Please, check the numeration of different figures!
The major imperfection and negative impression after reading this section is that the section contain very many citations taken from the literature. Over citation and constant comparison with the literature complicate understanding of the results and more important a clear differentiation between authors’ contribution from the current study and other studies. Therefore, I strongly suggest to authors to reconsider this part and elaborate much more on the description and present of their results and to provide a better discussion of the results.
Conclusion: It describes the results of the study. ‘Thermal treatment did not reduce the stability…’, which treatment – 45 and/or 85°C? Please, add in brackets.
I think that the paper is insufficient for publication in its present form, but it can be reconsider after revision.
Round 2
Reviewer 3 Report
To Authors:
A modified verion of manuscript already considered for publication in Molecules was re-submitted. I am pleased to note that the authors have taken into account suggestions and comments of me and another reviewer, and have made revision. I do not have further notes to the authors. Now, I think that the manuscript version is suitable for publication.